# Towards Multi-Functional SiO_2_@YAG:Ce Core–Shell Optical Nanoparticles for Solid State Lighting Applications

**DOI:** 10.3390/nano10010153

**Published:** 2020-01-16

**Authors:** Mahdi Kiani Khouzani, Abbas Bahrami, Maryam Yazdan Mehr, Willem Dirk van Driel, Guoqi Zhang

**Affiliations:** 1Department of Materials Engineering, Isfahan University of Technology, Isfahan 84156-83111, Iran; mahdi.kiani1996@gmail.com (M.K.K.); a.n.bahrami@gmail.com (A.B.); 2Faculty EEMCS, Delft University of Technology, Mekelweg 4, 2628 CD Delft, The Netherlands; willem.van.driel@signify.com (W.D.v.D.); g.q.zhang@tudelft.nl (G.Z.); 3Signify, High Tech Campus 48, 5600 JW Eindhoven, The Netherlands

**Keywords:** core–shell, nanoparticles, SiO_2_@YAG:Ce, Optical properties, Small angle X-ray scattering (SAXS), Solid state lighting

## Abstract

This paper aims to investigate the synthesis, structure, and optical properties of SiO_2_@YAG:Ce core–shell optical nanoparticles for solid state lighting applications. YAG:Ce phosphor is a key part in white light emitting diodes (LEDs), with its main functionality being the generation of yellow light. Generated yellow light from phosphor will be combined with blue light, emitted from chip, resulting in the generation of white light. Generated light in LEDs will often be scattered by SiO_2_ nanoparticles. SiO_2_ nanoparticles are often distributed within the optical window, aiming for a more homogeneous light output. The main idea in this research is to combine these functionalities in one core–shell particle, with its core being SiO_2_ and its shell being phosphor. In this study core–shell nanoparticles with different Ce^3+^ concentrations were synthesized by a sol–gel method. Synthesized nanoparticles were characterized by X-ray diffraction (XRD), small angle X-ray scattering (SAXS) analysis, high resolution transmission electron macroscopy (HRTEM), Fourier transform infrared (FTIR), and photoluminescence spectroscopy. Luminescence characteristics of SiO_2_@YAG:Ce core–shell particles were compared with that of SiO_2_/YAG:Ce mixture composite, which is now used in commercial LEDs. Obtained results showed that core–shell nanoparticles have comparatively much better optical properties, compared to SiO_2_/YAG:Ce mixture composite and can therefore be potentially used in LEDs.

## 1. Introduction

Light converting phosphors have a significant contribution in many photonic and optoelectronic devices and components [1,2,3,4,5]. Amongst different phosphors, yttrium aluminum garnets (Y_3_Al_5_O_12_: YAG), doped with rare-earth elements, i.e., Ce^3+^, are considered as one of the most widely used photonic materials [6] YAG:Ce phosphors have unique optical properties, making them excellent choices for different applications, ranging from lasers to solid state lighting sources [7,8,9,10]. This includes cathode-ray tubes (CRT), field emission displays (FED), scintillators, plasma display panels (PDP), electroluminescent displays, and light emitting diodes (LED). One of the most important characteristics of YAG:Ce phosphor is its ability in converting blue light radiation into a very wide band yellow emission. In LEDs, for instance, YAG:Ce phosphor absorbs the blue light and radiates yellow light. Generated yellow light will then be combined with non-absorbed blue light, resulting in the final emission of white light in LEDs [11]. This is schematically depicted in Figure 1. Currently, phosphor-based white LEDs are gaining significant attention because of their advantages over existing incandescent and halogen lamps such as their higher reliability, lower energy consumptions, and higher light output [7]. According to the US Department of Energy (DOE) report, LEDs are expected to represent 84% of lumen-hour sales for general illumination products by 2030 [12]. Based on the current state of technology, the report predicts that a 40% reduction in energy consumption in the lighting sector can be attained by 2030, if traditional light sources are replaced with LEDs [12]. Predicted energy savings are mostly in linear fixture, low/high bays, and outdoor applications. Overall, it appears that LEDs will have a prominent contribution to energy consumptions and energy savings in the lighting industry. In solid state lighting applications, and more specifically in LEDs, YAG phosphor particles play a vital role. The efficiency of an LED is directly linked to the light conversion efficiency of phosphors. In order to achieve a higher luminous efficiency and color homogeneity in LEDs, the so-called scattering enhancement particles (SEP), such as SiO_2_, TiO_2_, CaF_2_, and CaCO_3_ are routinely added to LED lens [13,14,15,16] (see Figure 1). Scattering agents are often mixed with phosphor particles, and the mixture composite, together with other additives, are added to the plastic lens. In this case, phosphor particles have optical functionality, while scattering agents obviously scatter the light to improve the color homogeneity of emitted light [14,17,18,19]. This paper investigates the idea of combining these functionalities in one multi-functional core–shell nanoparticle in which the shell is a phosphor layer and the core is the scattering agent. Luminescent core–shell structures appear to be very promising and are now becoming more popular in different applications [20,21,22,23,24,25,26]. Core–shell nanostructures show superior physical and chemical properties in comparison with their single-component counterparts, making them attractive choices for a wide range of applications not only in solid state lighting, but also in biomedicine, energy conversion, storage, and catalysis. In core–shell structures, shell often protects the core from the working environment, adds new chemical or physical capabilities to the core, maintains structural integrity, limits volume expansion, and inhibits particle aggregations [27,28]. Amongst different scattering agents in LEDs, silica is the most widely used one, due to the ease of synthesis, great controllability over the size, and structural stability. The focus in this study is to synthesize SiO_2_@YAG:Ce nanoparticles, using a sol–gel process, and to evaluate their optical properties. The aim is to achieve perfect spherical shaped core–shell particles, with the core being SiO_2_ and the shell being a homogenous thin phosphor shell. This ultimately ends up in multi-functional particles, which simultaneously generate and scatter light.

## 2. Materials and Methods

### 2.1. Synthesis of Core–Shell SiO_2_@YAG:Ce Nanoparticles

SiO_2_@YAG:Ce nanoparticles doped with 0.25, 0.5, 1, and 2 atomic weight % Ce were prepared by the sol–gel method [29,30,31,32]. Pure Y(NO_3_)_3_·6H_2_O (Sigma-Aldrich, St. Louis, MO, USA), Al(NO_3_)_3_·9H_2_O (Sigma-Aldrich, St. Louis, MO, USA), and Ce(NO_3_)_3_·6H_2_O (Sigma-Aldrich, St. Louis, MO, USA) were used as the cations’ sources. In the first stage, the nitrate salts were mixed in stoichiometric ratios (at a molar ratio Y:Al of 3:5) and dissolved in deionized water and citric acid (Merck, Darmstadt, Germany), such a way that the molar ratio CA:M^3+^ is 1:1 (M^3+^ = Y^3+^ + Al^3+^ + Ce^3+^). The solution was magnetically stirred until a clear solution was obtained. Then, ethylene glycol (Merck, Darmstadt, Germany), which acts as a polymerization agent, was added to the solution and the solution was heated with stirring at 80 °C for 30 min. pH was constantly monitored and kept at 3 by dropwise addition of NH_3_ solution (Merck, Darmstadt, Germany). Effects of variation in the pH between 3 and 4 on the structure of synthesized phosphor was also studied. In the second stage, SiO_2_ nanoparticles and polyethylene glycol (PEG; 10000-PEG, Merck, Darmstadt, Germany) as surfactant [20,22], were dispersed in deionized water. The SiO_2_/PEG-containing solution was added to the primary solution, while the solution was magnetically stirred. Figure 2 shows the X-ray diffraction (XRD) spectrum and FESEM image of SiO_2_ nanoparticles with mean particle size of 25 nm. SiO_2_ particles are spherical with a perfectly narrow size distribution. XRD spectrum also showed that SiO_2_ has an amorphous structure. To obtain core–shell nanopowders, the solution was further heated at 80 °C (for about 1.5 h) until it turned into a gel. The obtained gel was then dried at about 120 °C for one day, followed by calcination at 1000 °C for 1, 2.5, 5, and 10 h. An overview of the whole synthesis process is shown in Figure 3. In order to investigate the effects of shell thickness on the optical properties of nanoparticles, this process was conducted twice for some samples, hereafter named coat II. Samples, named coat I, are those that are coated one time. Optical properties of core–shell particles were compared with SiO_2_/YAG:Ce composite mixture (which is made of mixing SiO_2_ and YAG:Ce particles). This mixture powder is now currently used in the LED industry.

### 2.2. Characterization of Core–Shell Nanoparticles

The crystal structures of as-prepared SiO_2_@YAG:Ce core–shell and Y_3_Al_5_O_12_ phosphor particles were evaluated by X-ray diffraction (XRD) with CuK_α_ radiation (λ = 1.54 Å). Small angle X-ray diffraction (SAXS) analysis was also carried out at ESRF in Grenoble to study the structural properties and mean shell thickness in synthesized core–shell powders under various reaction conditions. The powders were supported by an adhesive tape. The scattering intensity *I*(*q*) was measured as a function of *q =* 4π/λ sin *θ*, with *2θ* being the scattering angle and λ = 0.154 nm being the wavelength of the used radiation. The surface morphology, the particle size and shape were analyzed with field emission scanning electron microscope (FESEM) and transmissions electron microscope (TEM/high resolution transmission electron macroscopy—HRTEM) (Tecnai 200STEM-FEG, FEI The Netherlands). Fourier transform infrared (FTIR) analyses were done by a Perkin Elmer Spectrometer (PerkinElmer, Waltham, MA, USA). Photoluminescence spectroscopy was carried out, using Avaspec 2048 TEC (Avantes, Apeldoorn, the Netherlands). Diffuse transmission spectra of core–shell particles were also measured.

## 3. Results and Discussion

### 3.1. Analyses of Synthesized Phosphor Particles

The XRD patterns of as-prepared phosphor powders, synthesized at different pH values and calcined at 1000 °C for 4 h, are shown in Figure 4. It appeared that the outcome of the synthesis was largely dependent on the pH values. The optimum synthesis condition was achieved when pH is strictly controlled at 3. Any deviation from pH = 3 towards higher values, and more specifically towards 3.5–4.0 pH range results in the appearance of YAM (Y_4_Al_2_O_9_) and YAP (YAlO_3_) transitional phases in as-synthesized powders. Figure 4 shows clear difference between XRD patterns of phosphor powders, synthesized at pH = 3 and that synthesized at pH range 3.5–4. Boukerika et al. [32] reported that pure cubic YAG phase with optimum optical properties can be attained at pH ≤4. It is reported that in case of pH ≥ 6, the formation of unfavorable phases such as Y_4_Al_2_O_9_ (YAM) and YAlO_3_ (YAP) is inevitable. The finding in this study is rather different with what is reported by Boukerika et al. [32], as the pure YAG was found out to be obtainable at pH ≤ 3. This possibly has to do with the fact that different raw materials are used in these studies. Both YAM and YAP phases are considered as impurities in YAG, as they cause energy level splitting of luminescence centers. This obviously adversely affects optical properties of phosphor. Overall, pH appears to be the most crucial controlling factor when it comes to the final phase composition of synthesized powders. Obtaining a pure homogenous phosphor phase is only possible when pH is strictly controlled. As mentioned earlier, the pH in this case was kept at 3 by dropwise addition of NH_3_ solution. Figure 4b also shows FESEM image of synthesized YAG:Ce powders. It is clear that synthesis has resulted in nanoparticles with spherical morphology with a perfectly homogenous dispersion. Elemental mapping of a synthesized YAG:Ce particle, depicted in Figure 5, shows that elements are also perfectly homogeneously distributed.

Figure 6a shows an example of the XRD pattern of samples annealed at 1000 °C. All peaks in the XRD patterns perfectly match with those of cubic YAG (JCPDS Card No. 79-1891) and no other crystalline phase such as YAlO_3_ (YAP) or Y_4_Al_12_O_9_ (YAM) can be detected. To make sure that synthesized YAG also has a structure, similar to that of industrial YAG, XRD spectra of synthesized and commercial YAG samples were compared (see Figure 6a). As mentioned earlier, single-phase pure YAG is very important in the luminescence efficiency of synthesized powders. Existing dopants obviously do not alter the structure of crystalline YAG. However, they surely change lattice parameters owing to the inequality of ionic radii between substituted yttrium ion and the dopant. A detailed XRD study on the effect of annealing time on the formation of YAG phase in core–shell nanoparticles was carried out. Figure 6b shows XRD patterns of SiO_2_@YAG:Ce nanoparticles after annealing from 1 to 10 h. The YAG phase starts to form after 1 h of annealing at 1000 °C. Therefore, one can conclude that crystallization time can be effectively decreased through the sol–gel method. In this work the minimum required crystallization time was found to be 1 h at 1000 °C. Appearance of sharp peaks and increasing the intensity of the main peak (2θ = 33.4, related to the crystallographic plane with Miller indices of {4 2 0}) is an indication of crystallization during annealing. The intensity of diffraction peaks of SiO_2_@YAG:Ce sample slightly increases with increase in the calcination time, inferring that the degree of crystallinity has increased. Meanwhile, this also indicates that mean crystallite size increases with increase in annealing temperature [30,31]. The same is expected when the annealing temperature is increased. Figure 6c shows schematic view of crystal structure of YAG, showing that YAG has a cubic garnet structure, containing octahedra (AlO_6_), tetrahedra (AlO_4_), and dodecahedra (YO_8_) with corner-shared O atoms. The co-doped Ce^3+^ ion as luminescence centers substitutes for Y^3+^ ion that is located in the position of YAG dodecahedral.

### 3.2. SAXS Analyses

Obtained SAXS data are presented in a log–log plot. SAXS curves are essentially plots of intensity as a function of the scattering vector *q*, which corresponds to the scattering angle *2θ*, given by [34]:(1)$$q=\left(\frac{4\pi }{\lambda }\right)\times \sin\theta .$$

The measured SAXS data were modeled using a commonly named global unified fit model [35,36]. In fact, this model includes a power-law regime in order to describe the mass or surface fractal and a Guinier regime to characterize the mean structural size, given by:(2)$$I\left(q\right)=\sum_{i=1}^{n}\left[G_{i}\exp(-q^{2}R_{gi}^{2}/3)+B_{i}\exp(-q^{2}R_{g\left(i+1\right)}^{2}/3)\times {\left\{\frac{q}{{\left[\mathrm{erf}\left(\frac{qR_{gi}}{\sqrt{6}}\right)\right]}^{3}}\right\}}^{-P_{i}}\right],$$
where, *R_gi_* is the radius of gyration, *erf* is error function, *i* refers to the differently sized structures, *G_i_* is the Guinier pre-factor, and *B_i_* is the pre-factor specific to the power-law scattering with an exponent *P_i_*. The mean primary particle size *d_p_* (for spherical particles) can be estimated from the radius of gyration *R_g_*, which can be obtained by Guinier’s law [35]:(3)$$d_{P}=2\sqrt{\frac{5}{3}}\times R_{g}.$$

In case nanoparticles have a log-normal size distribution, in order to characterize the particle size distribution from the SAXS data, three fitting parameters, *Rg*, *G*, and *B* are often used. The geometric standard deviation (*σ_g_*) in this case is given by [36]:(4)$$\sigma_{g}=\exp\left(\sqrt{\ln\frac{\left[B\times \frac{R_{g}^{4}}{1.62G}\right]}{12}}\right),$$
which characterizes the width of the size distribution. In the scattering vector *q* region, the scattering intensity *I*(*q*) can be characterized by the so-called power law [37]:(5)$$I\left(q\right)=B\times q^{-p},$$
where *B* is the power-law pre-factor and *P* refers to the power-law exponent. From the log *I*(*q*) versus log *q* curves and slopes of linear region (at large values of scattering vector *q*), the values of the exponent *P* can be measured. In order to study particle surface characteristics like roughness, the surface-fractal dimension or *d_s_* (*d_s_* = 6 − P) is often used [36,37]. This is in particular useful for ideal two-phase structure with smooth surfaces and sharp boundaries.

In order to determine the effective thickness of boundary layer along the radial direction of the sphere, the following formula is suggested [38,39]:(6)E=2√3σ,
where *σ* is the standard deviation of the Gaussian smoothing function and *E* is the thickness of the diffuse boundary layer, which can be measured by [38]:(7)$$I\left(q\right)\approx K_{p}\times q^{-4}\times \left(1-q^{2}E^{2}/12\right).$$

Figure 7a shows the scattering curves *I*(*q*) measured for pure SiO_2_ and SiO_2_@YAG:Ce core–shell nanoparticles. Before the coating was applied, the measured SAXS curve of un-coated SiO_2_ did not exhibit any specific side maxima. In order to extract structural information from the spectra, the data of the pure SiO_2_ were fitted by Guinier law (Equation (2)), the measured radius of gyration for primary particle of SiO_2_ (*R_G_*,_*P*_) was 8.9 nm (*d_p_* = 22.97 nm, *σ_g_* = 1.34), which is in accordance with TEM images. The power-law fit in the high of scattering vector *q* region follows Porod’s law, i.e., *I* ∼ *q*^−3.67^, implying that these silica nanoparticles have almost smooth surfaces (surface fractals or *D_s_* = 2.33). After preparation of core–shell nanoparticles, a specific side maximum or a shoulder appears in SAXS spectra (highlighted with red arrow). On the other hand, the coating process resulted in the formation of a YAG:Ce shell, growing on the SiO_2_ particle surface gradationally by heterogeneous nucleation. Similar side maximum or shoulders have been reported in the literature as particle−particle interactions (structure factor) [40,41,42,43]. In SiO_2_@YAG:Ce with two times coating (coat II), the progressive growth of shell is more visible in the SAXS in which the shoulder or specific side maximum is more clear (Figure 7b), suggesting that the shell grows with time. So, the YAG:Ce shell becomes thicker and the side maximum or shoulder is shifted to smaller scattering vector *q* region. In order to calculate the shell thickness, the SAXS data of the pure SiO_2_@YAG:Ce (coat I and II) were fitted by Equation (7), as displayed in Figure 7a and Figure 6b. The calculated mean thicknesses of the diffuse and formed interfacial boundary nanostructured layer of one and two cycle coating are 2.8 nm and 7.7 nm, respectively. The stability of SiO_2_@YAG:Ce core–shell nanoparticles at 1000 °C for different time was investigated by heating the sample from 1 to 10 hours in air. Figure 8 shows SAXS measurement of SiO_2_@YAG:Ce nanoparticles, calcined at 1000 °C for different times. It is noticeable that SAXS curves for all the specimens is almost similar, inferring that after calcination the core–shell structure of SiO_2_@YAG:Ce nanoparticles was perfectly preserved. Even heating the sample up to 10 h hardly results in any change in the morphology of nanoparticles, inferring that SiO_2_ cores are still encaged within the YAG:Ce shells. This obviously shows high thermal stability of SiO_2_@YAG:Ce core–shell nanoparticles [44,45]. It is noteworthy that the exponent of power-law fit at large *q* is greater than four (for example, I ∼ q^−4.20^), inferring that there exists a sprayed and formed boundary nanostructure like a thin layer formed on the particle surface of SiO_2_.

### 3.3. HRTEM/TEM Observations

HRTEM observations show that a relatively homogeneous thin layer of YAG:Ce has formed on the surface of the SiO_2_ particles (see Figure 9a). The size of this thin outer shell on the surface of particles was measured to be approximately 3 nm for one cycle coating, which is in a perfect agreement with SAXS calculations. Figure 9b shows TEM image of a single nanosphere after applying second layer of YAG:Ce coating. As can be seen, the thickness of the shell has increased from 3 to 7 nm. So, overall the thickness of the shell after one cycle coating was ∼3 nm and that increases to ∼7 nm after second cycle of coating. This is again in a good agreement with SAXS measurements. Figure 9c shows elemental mapping of alloying elements in synthesized particles. Elements are clearly homogeneously distributed in synthesized particles, inferring that YAG has been successfully synthesized on silica particles.

### 3.4. FTIR Analyses

Fourier transform infrared (FTIR) spectroscopy was used to study changes in the chemical structure of samples during synthesis (see Figure 10). Infrared spectra were recorded using a Perkin–Elmer Spectrum 100 series spectrometer in the attenuated total reflection (ATR) mode for 200 scans at a resolution of 4 cm^−1^. In the spectrum related to the SiO_2_ particles, the absorption bands due to OH (3435 cm^−1^), Si–O–Si (801 cm^−1^), Si–OH (950 cm^−1^), and SiO (471 cm^−1^) bonds were observed. Abovementioned absorption peaks show that SiO_2_ particles contain a large amounts of hydroxide (OH) groups and water (H_2_O) on the surface [46]. The SiOH groups play a very significant role for bonding the ions to the YAG:Ce shell. The FTIR spectrum, related to pure YAG:Ce powders, shows a strong absorption peak at 800 cm^−1^ and a weak peak at 455 cm^−1^, which are attributable to the absorption of AlO and YO bonds, respectively [32]. The dopant presence can also be confirmed by the peak at 516 cm^−1^, which is attributable to the vibration mode of the Ce–O bond. For the SiO_2_@YAG:Ce core–shell sample, the characteristic absorption peaks of the AlO bond (788 cm^−1^) for YAG:Ce and the Si–O–Si bond (1055 cm^−1^) for amorphous SiO_2_ are clearly visible. It appears that the weak signal of AlO bond has been affected and covered by the bending vibration of SiO bond at 471 cm^−1^. FTIR results are also in a perfect match with the XRD, HRTEM, and SAXS analyses, again confirming the formation of a crystallized YAG:Ce coatings on the silica surface via the sol–gel deposition.

### 3.5. Photoluminescence (PL) and Diffuse Transmission Spectra (DTS) Analyses

The photoluminescence (PL) spectra of YAG:Ce-coated silica nanoparticles as well as that of commercial and synthesized SiO_2_/YAG:Ce mixture samples are shown in Figure 11a. The emission spectra consist of a typical broad emission band centered at 540 nm and a shoulder at longer wavelength side. The former is assigned to the 5*d*_1_ → 2*f*_5/2_ and the latter to the 5*d*_1_ → 2*f*_7/2_ transition in Ce^3+^. The broad band spread from 500 to 650 nm is an ideal yellow light emission, which is in association with blue light emitted by InGaN LED chips. It is known that Ce^3+^ ion with a 4*f*_1_ electron configuration has two ground states of 2*f*_5/2_ and 2*f*_7/2_ owing to the spin-orbit coupling [47]. It is also noticeable that peak positions of YAG:Ce samples are not affected by SiO_2_ nanoparticle. The observed peak is due to characteristic transition of Ce^3+^. The fact that peak position is not changed in core–shell samples infers that the presence of SiO_2_ as the core hardly affects the structure and luminescence properties of YAG:Ce phosphor. Figure 11a also depicts that the higher the thickness of YAG:Ce shell, the higher is the maximum intensity in the PL spectrum. As can be seen, SiO_2_@YAG:Ce (1.0 at.% Ce) nanoparticles show the highest emission, compared to commercial and synthesized SiO_2_/YAG:Ce mixture composite, that is in agreement with similar works in this field [20,21,22,48,49,50,51]. The increased PL intensity of YAG:Ce in the core–shell state is believed to result from the inhibition of surface states in YAG:Ce nanoparticles and the higher light extraction at the SiO_2_/YAG:Ce interface. Emission intensity improvement of core–shell nanoparticles can also be attributed to the fact that a significant amount of non-radiative centers exist on the surface of core SiO_2_ are decreased by the coating effect of the YAG:Ce shell (as a shielding layer). Obviously, the surface –OH groups play a crucial role in the PL quenching. Surface elimination of –OH groups due to the surface coating originates from hydrogen bonds on the surface of SiO_2_ (see Figure 10). This will certainly influence the radiative relaxation pathway. Overall, it appears that using SiO_2_ nanoparticle as cores significantly enhances the emission intensities of core–shell particles due to light absorption/scattering by coated nanoparticle [20]. The number of coating cycles plays a vital role in enhancing the PL intensity of the core–shell particles (Figure 11a). The increase in the PL intensity with double coating cycle is obviously attributable to the increase in the thickness YAG:Ce shell on the SiO_2_ cores, which in turn increases emitting ions (Ce^3+^) per core–shell particle [22]. Figure 11b shows the effects of Ce content on the emission peak intensity of SiO_2_@YAG:Ce nanoparticles. SiO_2_@YAG:Ce nanoparticles show the highest emission when doped with 1.0 at.% Ce, compared with SiO_2_/YAG:Ce (commercial and synthesized mixture composite), which is in agreement with previous investigations [30,52,53,54]. The enhancement of PL intensity with an increase in Ce^3+^ concentration is due to the efficient incorporation of Ce as luminescence centers into the host material; in this case being YAG. The luminescent intensity is to a large extent affected by the average distance between luminescent centers. The distance among luminescence centers or active ions decreases due to increase in the concentration of doped ion to values more than 1.0 at.%. Further increase in Ce^3^^+^ dopant concentration results in an increase in the unit cell parameters, which may in turn enlarge the distance between Ce^3^^+^ ions in the YAG:Ce structure. More non-radiation can also be the result of cross relaxation of the excessive dopant (Ce^3+^ ions). Given that the ionic radius of cerium is approximately 17.5% larger than that of yttrium ion, the substitution of Ce^3+^ ion to Y^3+^ sites is rather difficult. The excessive doping hinders the substitution of Y^3^^+^ with Ce^3^^+^ ions. Thus, the incorporation of high concentration of Ce^3^^+^ shortens the average distance between Ce^3^^+^ and non-radiative transitions possibility increases, which leads to the concentration quenching of Ce^3+^ ions [53,54]. The other reason for the reduction of emission intensities of YAG with high Ce^3+^ concentration is the sectional absorption of excitation photons by YAG as host material. This induces less excitation of Ce^3+^ ion and thus the PL intensity of the YAG:Ce phosphors is lowered [53]. At last, cerium ion oxidation (Ce^3+^ → Ce^4+^), i.e., CeO_2_, which expectedly takes place close to the surface can decrease the photoluminescence intensity of the Ce^3+^ luminescence centers [54]. In order to avoid abovementioned problems and achieve the highest brightness and efficiency in YAG with Ce as luminescence centers, achieving a homogeneous distribution of Ce^3+^ ions in the YAG host is extremely important. It appears that synthesized SiO_2_@YAG:Ce core–shell particles have a comparatively higher PL intensity, compared to SiO_2_/YAG:Ce mixture sample, which obviously has to do with optimum distribution of Ce^3+^ ions over the surface of SiO_2_ particles. More importantly, implementing this methodology is associated with partial substitution of precious lanthanide elements (heavy rare earth) by inexpensive SiO_2_ particles, which has major positive implications, when the final price of the product is concerned.

Diffuse transmission spectra of YAG:Ce and SiO_2_@YAG:Ce particles are shown in Figure 12. The low transparency of YAG:Ce nanoparticles at around 350 and 450 nm can be ascribed to the intrinsic absorption of Ce^3+^ ions that are caused by 4*f*−5*d*_2_ and 4*f*−5*d*_1_ transitions, respectively. It is clear that the absorption of YAG:Ce is not affected by SiO_2_. The relative intensity decrease in the absorption peak of the core–shell SiO_2_@YAG:Ce nanoparticles is possibly due to presence of SiO_2_ particle that could change the dielectric constant (as a complete transparent materials) inside YAG:Ce shell and thus decrease the absorption intensities [55]. It is seen that the amount of SiO_2_ nanoparticle in the core–shell system is not high enough to decrease the transmission of SiO_2_@YAG:Ce. Figure 12b shows diffuse transmission spectra (DTS) of YAG:Ce-coated Silica nanoparticles as well as that of commercial and synthesized SiO_2_/YAG:Ce samples. It is noticeable that relative transmission increased in core–shell nanoparticles and also increased with two cycle coating. According to Mie’s theory, scattering is caused by difference in refractive indexes between scattering agents and surrounding media [56,57]. The refractive indexes of SiO_2_ and YAG:Ce are about 1.46 and 1.82, respectively. It is clear that in core–shell nanoparticles scattering process of incident light is more efficient and this in turn has increased transmission.

## 4. Conclusions

This paper investigated the synthesis, structure, and optical properties of multi-functional SiO_2_@YAG:Ce nanoparticles for solid state lighting applications. The following conclusions can be drawn:
-Results showed that the final phase composition of synthesized powders largely depended on pH values. The optimum condition was achieved when pH was strictly controlled at pH = 3. Any deviation from pH = 3 towards higher values, and more specifically towards the 3.5–4.0 pH range resulted in the appearance of YAM (Y_4_Al_2_O_9_) and YAP (YAlO_3_) transitional phases in as-synthesized powders. Both phases are known to have adverse attribution to the optical characteristics of YAG:Ce powders.-SAXS analysis showed that the mean thickness of YAG:Ce shell after one and two coating cycles were 2.8 nm and 7.7 nm. This was in agreement in HRTEM observations.-Heating the sample up to 10 h hardly resulted in any change in the morphology of nanoparticles, inferring that SiO_2_ cores were still encapsulated by YAG:Ce shell, obviously showing perfect thermal stability of SiO_2_@YAG:Ce core–shell nanoparticles.-SiO_2_@YAG:Ce (1.0 at.% Ce) core–shell nanoparticles show the highest emission, compared to commercial and synthesized SiO_2_/YAG:Ce mixture composite.-The number of coating cycles played a vital role in enhancing the PL intensity of the core–shell particles. The increase in the PL intensity with double coating cycle was obviously attributable to the increase of the shell thickness (YAG:Ce) on the SiO_2_ cores, which in turn increased emitting ions (Ce^3+^) per core–shell particle.

## Figures and Tables

**Figure 1 nanomaterials-10-00153-f001:**
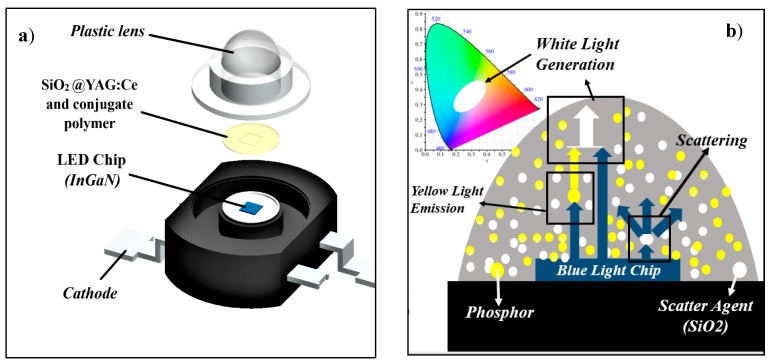
(**a**) Schematics of light emitting diode (LED) structure and (**b**) mechanism of white light generation.

**Figure 2 nanomaterials-10-00153-f002:**
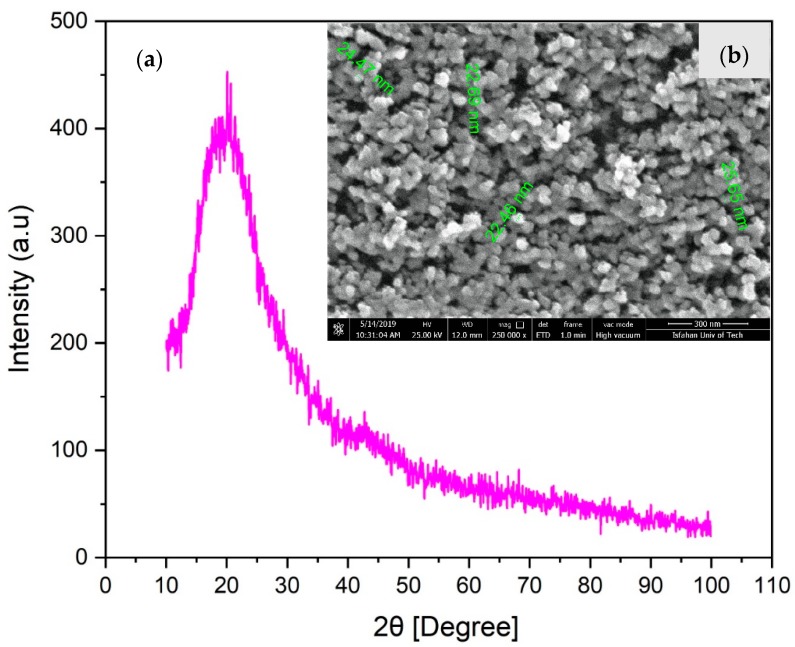
(**a**) XRD spectrum and (**b**) FESEM image of nano-sized SiO_2_ particles.

**Figure 3 nanomaterials-10-00153-f003:**
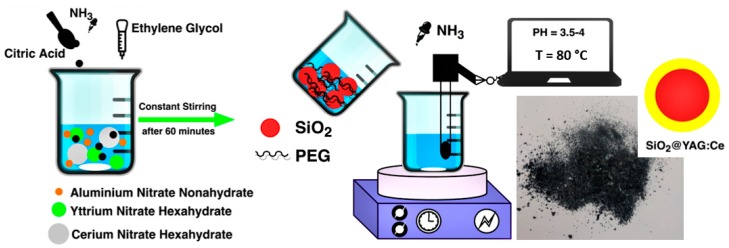
An overview of the synthesis process.

**Figure 4 nanomaterials-10-00153-f004:**
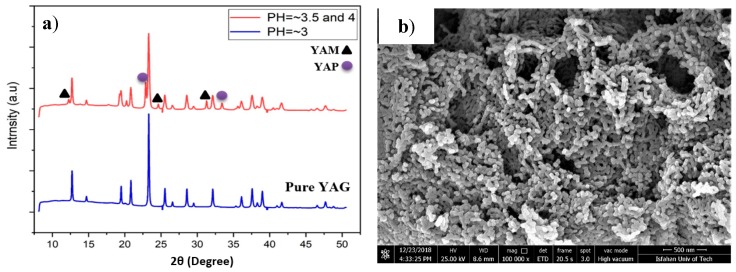
(**a**) XRD patterns of YAG:Ce nanoparticles, synthesized using solutions with different pH values and (**b**) FESEM image of synthesized YAG:Ce powders.

**Figure 5 nanomaterials-10-00153-f005:**
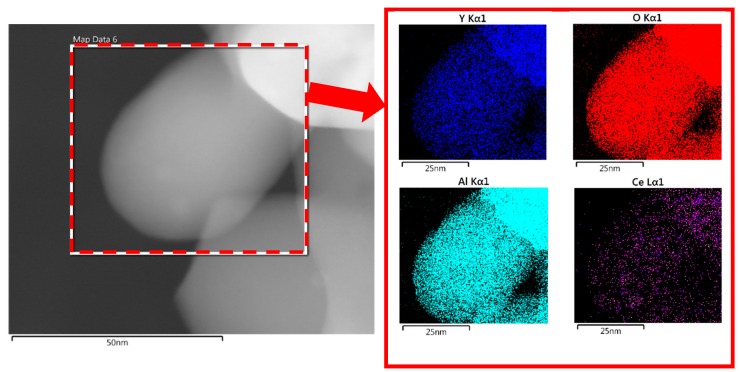
Elemental mapping of a YAG:Ce nanoparticle.

**Figure 6 nanomaterials-10-00153-f006:**
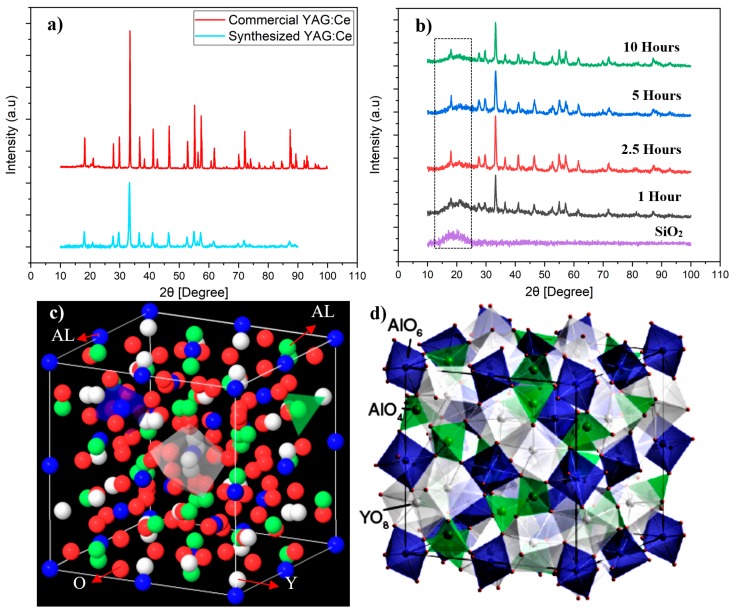
(**a**) X-ray diffraction patterns of synthesized YAG:Ce nanoparticles and commercial YAG:Ce, (**b**) X-ray diffraction patterns of SiO_2_@YAG:Ce nanoparticles after annealing, and (**c**,**d**) crystal structure of YAG [33].

**Figure 7 nanomaterials-10-00153-f007:**
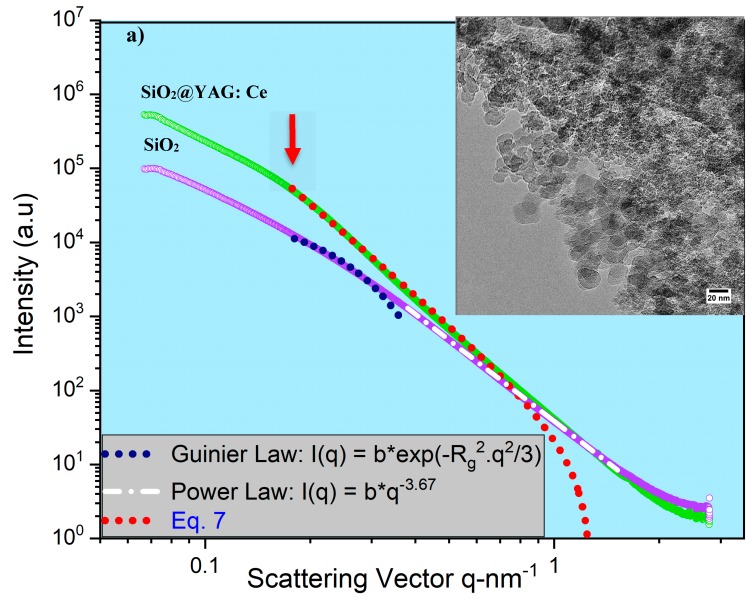
(**a**) Small angle X-ray diffraction (SAXS) curves of SiO_2_ and SiO_2_@YAG:Ce and (**b**) SAXS curves of SiO_2_ and SiO_2_@YAG:Ce (coat I and II).

**Figure 8 nanomaterials-10-00153-f008:**
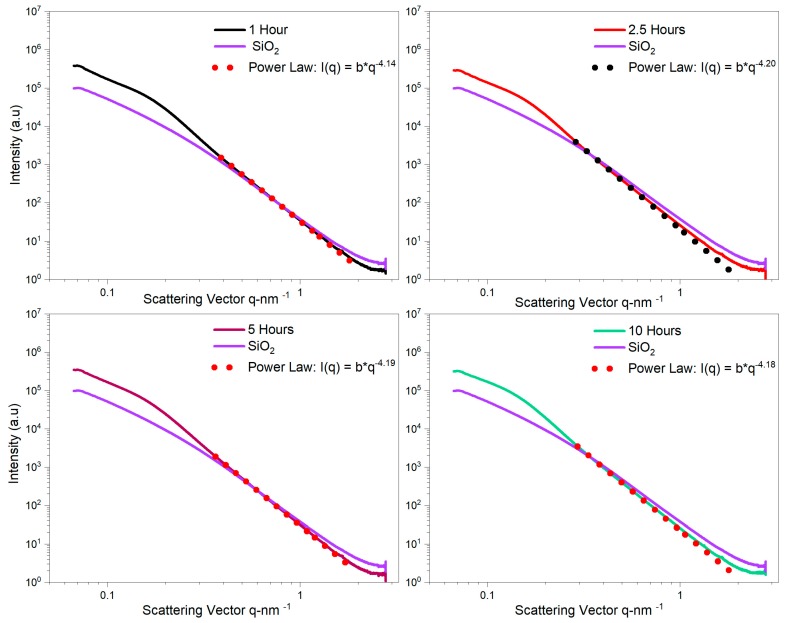
SAXS curves of core shell nanoparticle calcined at 1000 °C for different times.

**Figure 9 nanomaterials-10-00153-f009:**
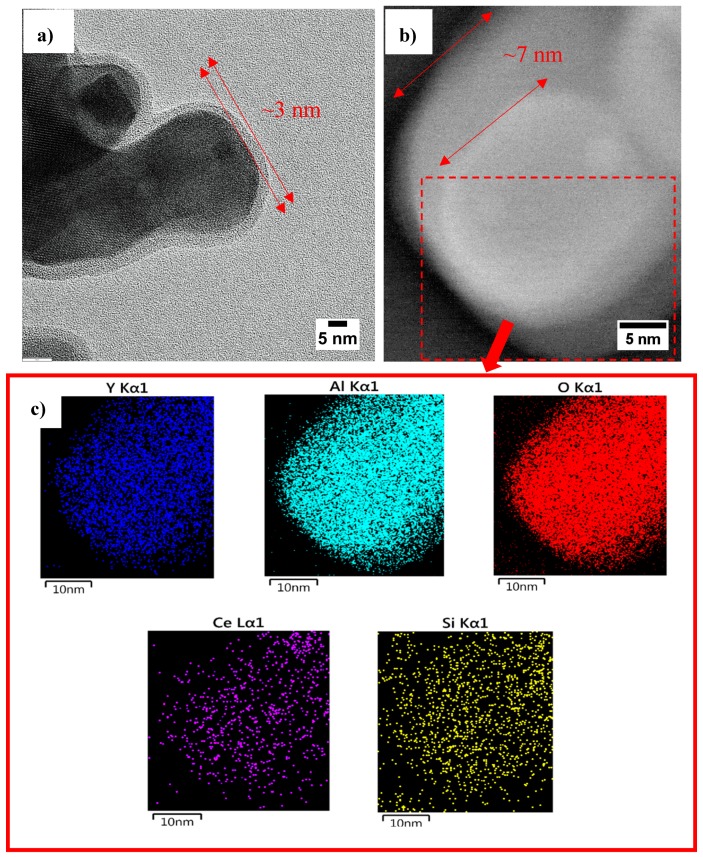
(**a**) High resolution transmission electron macroscopy (HRTEM) image of SiO_2_@YAG:Ce (coat I) and (**b**) TEM image of SiO_2_@YAG Ce (coat II), and (**c**) elemental mapping analysis.

**Figure 10 nanomaterials-10-00153-f010:**
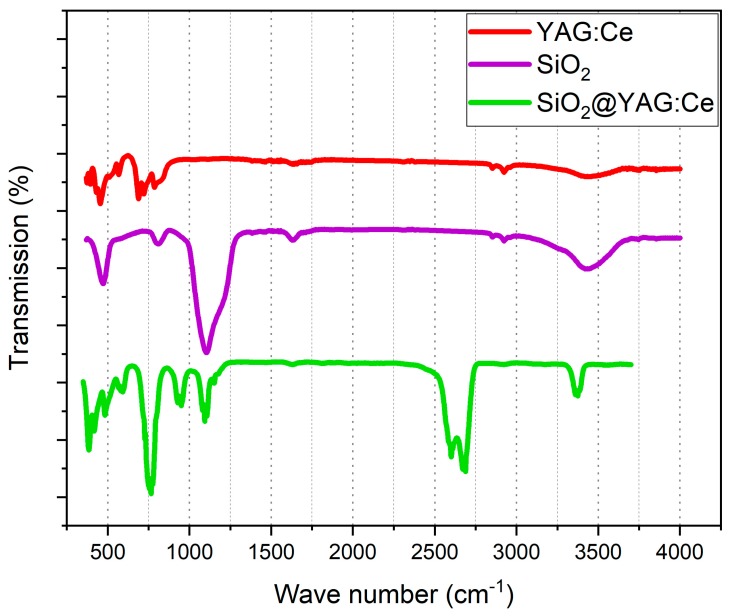
FTIR spectra of SiO_2_ spheres, SiO_2_@YAG:Ce core shell phosphors, and that of pure YAG:Ce phosphors.

**Figure 11 nanomaterials-10-00153-f011:**
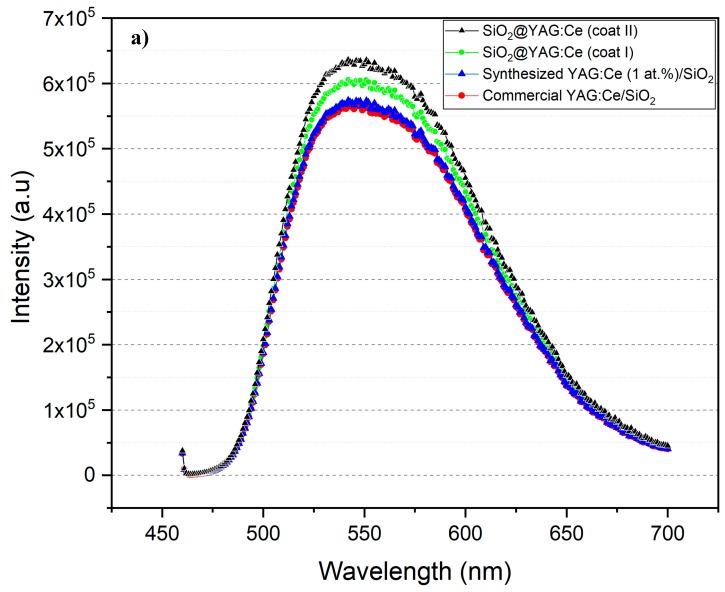
(**a**) Photoluminescence emission spectra of SiO_2_/YAG:Ce (commercial and synthesized) composite and SiO_2_@YAG:Ce core–shell particles and (**b**) luminescent main intensity of SiO_2_@YAG:Ce varying with the Ce concentrations and SiO_2_/YAG:Ce (commercial and synthesized composite).

**Figure 12 nanomaterials-10-00153-f012:**
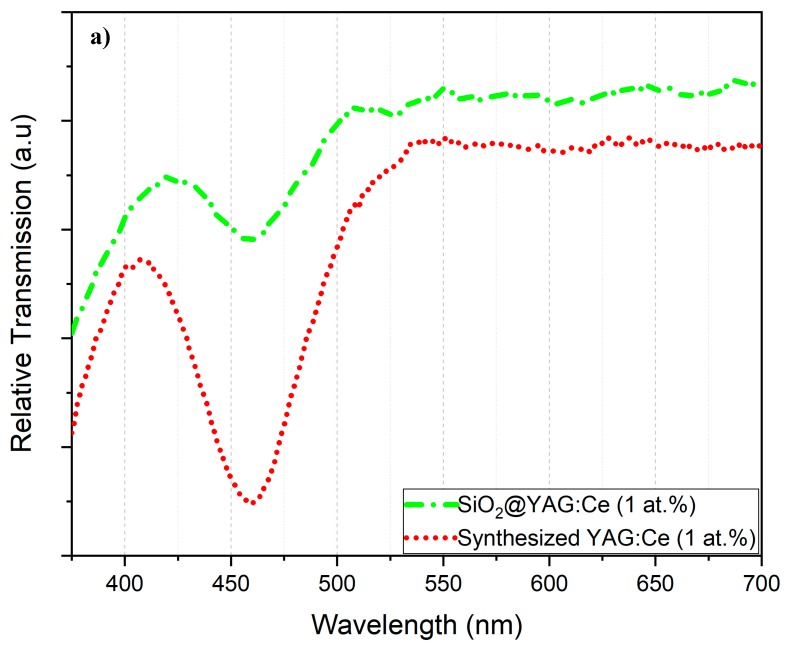
Diffuse transmission spectra of (**a**) YAG:Ce and SiO_2_@YAG:Ce and (**b**) synthesized SiO_2_/YAG:Ce (1 at.%) composite and SiO_2_@YAG:Ce core–shell nanoparticles.

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
