# Peer review of "Towards Multi-Functional SiO2@YAG:Ce Core–Shell Optical Nanoparticles for Solid State Lighting Applications"

_nanomaterials, 2020, doi:10.3390/nano10010153_

Round 1

Reviewer 1 Report

The author demonstrated core-shell structured nanoparticle using SiO2 and YAG:Ce.

However, it is hard to evaluate.

How much is the demonstrated result valuable?

The result should be compared with that of other literature.

It seems to have ~18% improvement (in figure 11a), and ~7% improvement (in figure 11b). 

Are they high enough? (should be compared with reference papers.)

Other issues

Figure 12 does not show the numerical value. So, the presented data cannot be compared with each other. considering the RI difference of YAG:[email protected] the scattering efficiency analysis would be useful. considering the scatter effect, total flux or viewing angle measurement is required.

Minor Issue

Expression error: "YAG:[email protected]" should be corrected with "[email protected]:Ce".

Author Response

Dear Reviewer, 

Thanks very much indeed for your comments. Here please find detailed response to your comments. I hope you find these answers convincing. 

Thanks and regards,

However, it is hard to evaluate. How much is the demonstrated result valuable? The result should be compared with that of other literature. It seems to have ~18% improvement (in figure 11a), and ~7% improvement (in figure 11b). 

Are they high enough? (should be compared with reference papers.)

Thanks for your comment. LED is now considered as a mature technology. A lot has been done in terms of materials and design optimizations. So, any improvement in the light output in LEDs will be marginal. However, given that a huge share of worldwide energy consumption is in the lighting sector belongs to LEDs, any improvement in the efficiency of LED sources has major positive implications for the worldwide energy consumption, environment and the social sustainability. Based on current trends, 10% improvement in the efficiency of LEDs means that up to 1000 million less LED stocks will be needed for lighting in 2035, which is obviously a huge number. So, in that sense 7% percent is certainly a major achievement. Speaking of comparing results with literature, I should mention that the phosphor/SiO2 mixture we used in this study is the composition currently being used by Philips lighting. This means that reported improvement is in fact improvement compared to the reference industrial sample. We have recently published a review in International Materials Reviews on the "degradation of optical materials in solid state lighting applications" in which lots of information is given about numbers and trends in LED industry. 

Other issues

Figure 12 does not show the numerical value. So, the presented data cannot be compared with each other. considering the RI difference of YAG:[email protected] the scattering efficiency analysis would be useful. considering the scatter effect, total flux or viewing angle measurement is required.

Minor Issue

Expression error: "YAG:[email protected]" should be corrected with "[email protected]:Ce".

The manuscript is modified.

Reviewer 2 Report

In this paper, Prof. M. Yazdan Mehr and co-workers report the syntheses and properties of core-shell nanoparticles of [email protected]:Ce synthesized by sol-gel method for photonic application. Although this report is of interest to me, the authors should revise the following points before publication.

[1] I think that the English language needs major corrections for the readers. For example, “This papers” in Line 357 of Page 16 should be corrected as “This paper”. Checking of the English language should be undertaken by negative speakers.

[2] The authors should write correctly the technical terms of “SiO2”, “Ce3+” and so on as the superscript and subscript. 

[3] The authors should provide the size and size distribution of core-shell [email protected]:Ce nanoparticles.

[4] The authors should analyze the molar percentage of Ce containing in the nanoparticles by ICP measurement. 

[5] Are the description of equation of (3), (4), (5) and (7) correct?

[6] Figure 12; If the vertical axis is “Relative Transmission”, the unit should be “arbitrary unit”, but not “%”.

Author Response

Dear Reviewer, 

Thanks so much for your valuable comments. We tried to address all comments and take all your recommendations into considerations. Here please find detailed response to each of your comments. Hope you find them satisfactory. 

Thanks and regards,

[1] I think that the English language needs major corrections for the readers. For example, “This papers” in Line 357 of Page 16 should be corrected as “This paper”. Checking of the English language should be undertaken by negative speakers.

As per your comment, English was thoroughly revised. 

[2] The authors should write correctly the technical terms of “SiO2”, “Ce3+” and so on as the superscript and subscript. 

As per your comment, the manuscript was amended. 

[3] The authors should provide the size and size distribution of core-shell [email protected]:Ce nanoparticles.

Thanks for your comment. In this study, SiO2 particles have mean size of 25 nm. They do have a very narrow size distribution, with the variation being in the order of maximum 5%. Also, synthesized phosphor particles have a very narrow size distribution (see Figure 4b). 

[4] The authors should analyze the molar percentage of Ce containing in the nanoparticles by ICP measurement. 

As you know, ICP of phosphors is a very complicated and difficult analysis on the grounds that phosphor solubility in acids is very poor. That makes this analysis very difficult. We are not concerned about the composition of Ce in the final powder. This is because we measured initial powders with 10-4 accuracy balance. Also, we are certain that there is no loss during synthesis. Moreover, the core idea of this research was to come up with the know-how of synthesizing core-shell particles. In that regards, marginal variations in Ce the chance is remote though) does not have a massive influence on the final results. 

[5] Are the description of equation of (3), (4), (5) and (7) correct?

Yes, they are correct. 

[6] Figure 12; If the vertical axis is “Relative Transmission”, the unit should be “arbitrary unit”, but not “%”.

You are absolutely right. This is changed in the manuscript. 

Round 2

Reviewer 2 Report

I think that the revised manuscript by Prof. M. Yazdan Mehr and co-workers is recommended for acceptance of the publication in Nanomaterials.

Author Response

Dear Reviewer,

Many thanks for your useful inputs and comments.

Regards,

Maryam Yazdan Mehr